Corrected: Publisher Correction

# A TonB-dependent receptor constitutes the outer membrane transport system for a lignin-derived aromatic compound

Masaya Fujita [1], Kosuke Mori[1], Hirofumi Hara[2], Shojiro Hishiyama[3], Naofumi Kamimura [1] & Eiji Masai [1]*

TonB-dependent receptors (TBDRs) mediate substrate-specific transport across the outer membrane, utilizing energy derived from the proton motive force transmitted from the TonB—ExbB—ExbD complex located in the inner membrane (TonB system). Although a number of TonB systems involved in the uptake of siderophores, vitamin B12 and saccharides have been identified, their involvement in the uptake and catabolism of aromatic compounds was previously unknown. Here, we show that the outer membrane transport of a biphenyl compound derived from lignin is mediated by the TonB system in a Gram-negative bacterium capable of degrading lignin-derived aromatic compounds, *Sphingobium* sp. strain SYK-6. Furthermore, we found that overexpression of the corresponding TBDR gene enhanced the uptake of this biphenyl compound, contributing to the improved rate of platform chemical production. Our results will provide an important basis for establishing engineered strains optimized for use in lignin valorisation.

---

[1] Department of Bioengineering, Nagaoka University of Technology, Nagaoka, Niigata, Japan. [2] Department of Chemical Process Engineering, Malaysia-Japan International Institute of Technology, Universiti Teknologi Malaysia, Kuala Lumpur, Malaysia. [3] Forestry and Forest Products Research Institute, Tsukuba, Ibaraki, Japan. *email: emasai@vos.nagaokaut.ac.jp

The cell envelope of Gram-negative bacteria comprises outer and inner membranes; therefore, nutrients must be passed through these two membranes to utilize them in the bacterial cells. In the outer membrane, transport of nutrients is achieved by passive transporters, such as porins and substrate-specific channels, and active transporters, TonB-dependent receptors (TBDRs)[1]. TBDRs mediate substrate-specific transport across the outer membrane, utilizing energy derived from the proton motive force transmitted from the TonB−ExbB−ExbD complex (TonB complex) located in the inner membrane (TonB system)[2,3]. A number of TonB systems involved in the uptake of siderophores, vitamin B12 and saccharides have been identified[4–6]. Although there are examples where the TonB system transports certain siderophores containing aromatic groups[7,8], its involvement in the uptake and catabolism of aromatic compounds was previously unknown. It was envisaged that TBDRs would have more diverse functions than is currently known based on the fact that many Gram-negative bacteria have a large number of TBDR-like genes in their genomes, whose functions are unknown[9,10].

Bacterial outer membrane transporters of aromatic compounds reported thus far have included passive transporters such as OpdK and OmpW, which are a vanillate-specific channel of *Pseudomonas aeruginosa* PAO1 and a naphthalene porin of *Pseudomonas fluorescens*, respectively[11,12]. Recently, novel functions of the active transporters, TonB-dependent receptors, have been documented, such as in the uptake of rare earth metals, membrane homeostasis, and secretion of proteins[13–15]. In addition, the upregulation of particular TBDR-like genes has been observed in *Sphingomonas wittichii* RW1 and a *Pseudomonas* strain during their growth in the presence of dioxin and vanillin, respectively, evoking the possibility that TBDRs participate in the outer membrane transport and catabolism of aromatic compounds[16,17]. Generally, TBDRs have a membrane-spanning barrel domain consisting of 22 antiparallel β-strands and an N-terminus region containing a short, conserved motif called the TonB box, which is essential for the interaction between TBDRs and the C-terminal region of TonB[2,18]. The inside of the β-barrel domain is filled with a plug domain that prevents nonspecific-substrate influx.

Lignin, a major component of plant cell walls, is the most abundant aromatic compound on Earth; thus, its industrial use is eagerly anticipated to achieve sustainable development. However, the effective utilization of lignin has not yet been established, mainly due to its structural complexity and recalcitrance[19]. Recently, the production of value-added chemicals from lignin through a combination of the chemical depolymerization of lignin and subsequent microbial conversion of the resulting low-molecular-weight aromatic compounds, known as 'biological funneling', has attracted considerable attention[20,21]. In nature, the biological degradation of lignin is thought to progress mainly via depolymerization by fungi and subsequent mineralization by bacteria of the resulting low-molecular-weight aromatic compounds[22]. Therefore, the elucidation of microbial lignin catabolism systems is extremely important not only for their application to microbial conversion of lignin into value-added chemicals but also for understanding the carbon cycle on Earth. *Sphingobium* sp. SYK-6 (a Gram-negative alphaproteobacterium) is the microorganism in which the catabolism of lignin-derived aromatic compounds has been best-characterized[20,23]. SYK-6 generates 2-pyrone-4,6-dicarboxylate (PDC), a promising platform chemical for the synthesis of highly functional polymers, as an intermediate metabolite in the catabolism of lignin-derived biaryls and monoaryls[23–25]. To date, a considerable portion of the catabolism of lignin-derived aromatic compounds in SYK-6 has been genetically and biochemically characterized. By using SYK-6 catabolic genes, PDC production from lignin-derived

aromatic compounds has been achieved[26–28]. However, little is known about how these various lignin-derived aromatic compounds are transported into the cell across the outer and inner membranes. Recently, we identified the major facilitator super-family transporter genes *pcaK* and *ddvK*, which are involved in the inner membrane transport of protocatechuate (PCA) and 5,5′-dehydrodivanillate (DDVA), respectively, in SYK-6[29,30]. Overexpression of *pcaK* and *ddvK* in a PDC-accumulating mutant of SYK-6 improved the rate of growth on substrates, substrate conversion and PDC production. Therefore, increased transporter gene expression should be able to enhance the production of metabolites from lignin derivatives. In contrast, the outer membrane transport of lignin-derived aromatic compounds, the initial stage of catabolism, remains largely unknown. In the SYK-6 genome (accession numbers AP012222 and AP012223), only one gene showed any similarity with known aromatic-compound porins, whereas there are 74 putative TBDR genes. Based on the fact that SYK-6 specializes in the degradation of lignin-derived aromatic compounds, we predicted the involvement of TBDR genes in the outer membrane transport of aromatic compounds[23].

Here, we show that the outer membrane transport of a biphenyl compound derived from lignin is mediated by the TonB system in SYK-6. To our knowledge, this is the first report experimentally demonstrating the involvement of the TonB system in the outer membrane transport and catabolism of an aromatic compound. Furthermore, we found that overexpression of the corresponding TBDR gene enhanced the uptake of this biphenyl compound, thus contributing to the improved rate of platform chemical production from lignin-derived compounds.

## Results

**TBDR genes induced by lignin-derived aromatic compounds.** SYK-6 has an *ompW*-like gene (SLG_38320), which has similarities with aromatic-compound porins reported in other bacteria. However, disruption of *ompW* did not affect the growth of SYK-6 on lignin-derived aromatic compounds (Supplementary Figs. 1–3). A phylogenetic tree was constructed based on the deduced amino acid sequence similarity between the 74 putative TBDR genes in SYK-6 and known TBDR genes (Supplementary Table 1). This analysis showed that 25 and 28 SYK-6 TBDRs could be classified into two specific clades separated from those including the known TBDRs (Fig. 1a). This fact suggests that these TBDRs may have unknown functions.

The expression patterns of all TBDR genes in SYK-6 cells incubated in Wx minimal medium containing SEMP (10 mM sucrose, 10 mM glutamate, 20 mg l$^{-1}$ methionine and 10 mM proline) in the presence and absence of lignin-derived aromatic compounds were investigated by DNA microarray analysis. We found that the transcription of 17 TBDR genes was specifically induced and increased 2–35-fold during growth with lignin-derived aromatic compounds, including guaiacylglycerol-β-guaiacyl ether (GGE; β-O-4 type), dehydrodiconiferyl alcohol (DCA; β-5 type) and DDVA (5-5 type) (Fig. 1b, Supplementary Table 2). Among these genes, the expression of SLG_07650 was induced 4.87-fold during growth with DDVA. The SLG_07650 gene is located upstream of the DDVA catabolic genes cluster, which includes genes for catabolic enzymes (*ligXa*, *ligZ* and *ligY*), an inner membrane transporter (*ddvK*), and a MarR-type transcriptional regulator (*ddvR*) that negatively regulates *ligXa* expression (Fig. 2a, b)[30]. The BOCTOPUS2 program predicted that the gene product of SLG_07650 forms a membrane-spanning barrel domain, comprising 22 antiparallel β-strands, which is a typical feature of TBDRs (Supplementary Fig. 4). In addition, the presence of an N-terminal signal sequence and a subsequent plug

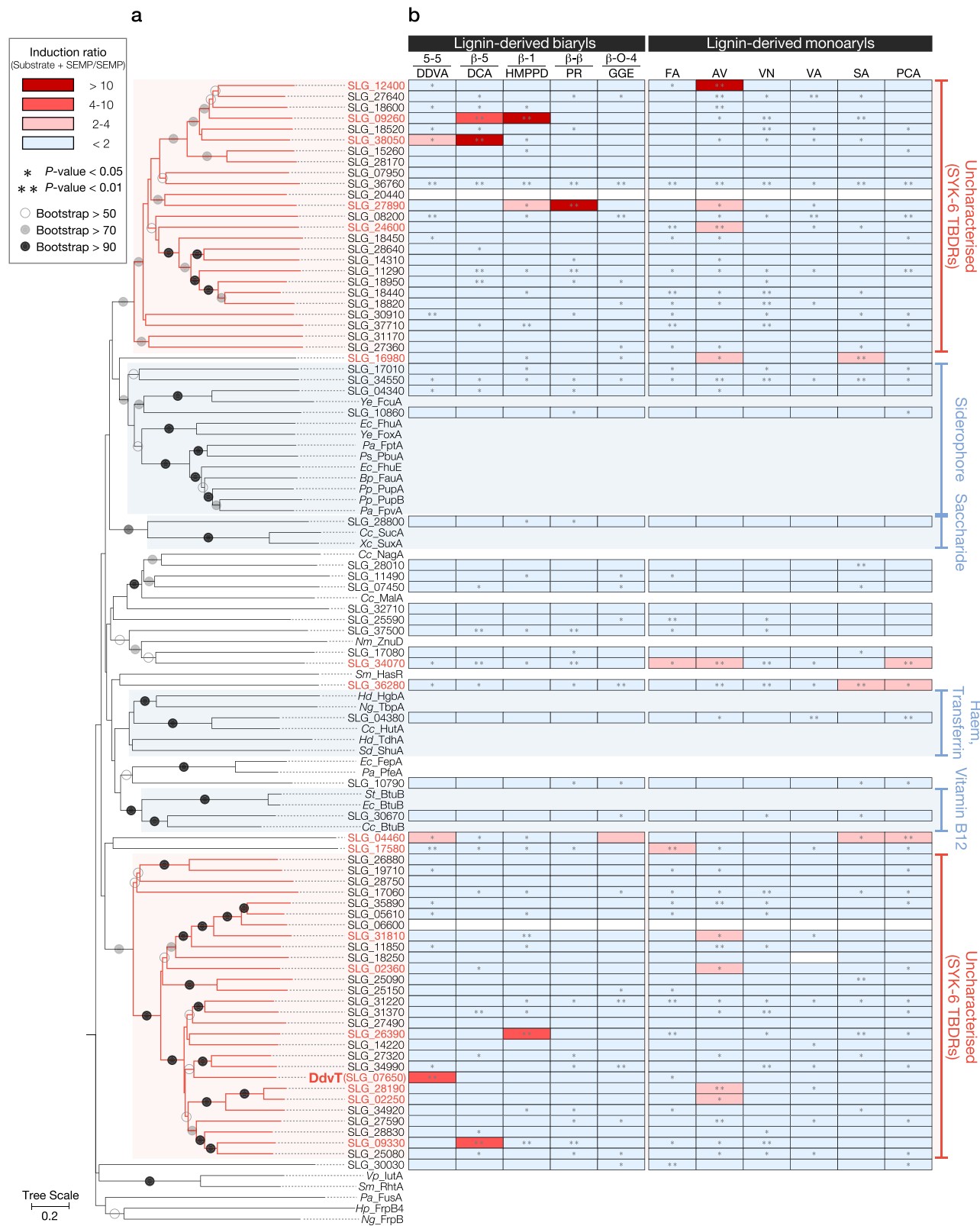

COMMUNICATIONS BIOLOGY | (2019) 2:432 | https://doi.org/10.1038/s42003-019-0676-z | www.nature.com/commsbio

domain-like sequence were predicted. We hypothesized that SLG_07650 is involved in the uptake of DDVA, and designated this gene *ddvT*.

**DdvT mediates the outer membrane transport of DDVA.** We performed western blot analysis using anti-DdvT antibodies against total membrane fractions obtained from SYK-6, *ddvT* mutant (Δ*ddvT*, Supplementary Fig. 2), and *ddvR* mutant (Δ*ddvR*) cells grown with or without 1 mM DDVA. DdvT was not detected in Δ*ddvT* cells, whereas production of DdvT in the wild type was highly induced during growth with DDVA (Fig. 2c, Supplementary Fig. 15a). In Δ*ddvR* cells, DdvT was produced in almost equal quantities to the wild-type cells incubated with

**Fig. 1** *Sphingobium* sp. strain SYK-6 has a number of TBDR genes of unknown function which are specifically induced in the presence of lignin-derived aromatic compounds. **a** A phylogenetic tree based on the alignment of the deduced amino acid sequences of the 74 putative TBDR genes in SYK-6 with those of the TBDR genes of known function (Supplementary Table 1). Bootstrap values (based on 1000 replicates) are shown at the branch points. **b** A heat map showing the fold change in the expression of each TBDR gene in cells grown in Wx minimal medium containing SEMP[37] with lignin-derived aromatic compounds versus those grown without lignin-derived aromatic compounds, as determined by DNA microarray analysis. Lignin-derived aromatic compounds used for the DNA microarray analysis were: DDVA, 5,5′-dehydrodivanillate (5-5 type); DCA, dehydrodiconiferyl alcohol (β-5 type); HMPPD, 1,2-bis(4-hydroxy-3-methoxyphenyl)-propane-1,3-diol (β-1 type); PR, pinoresinol (β-β type); GGE, guaiacylglycerol-β-guaiacyl ether (β-O-4 type); FA, ferulate; AV, acetovanillone; VN, vanillin; VA, vanillate; SA, syringate; PCA, protocatechuate. The chemical structures of these compounds are shown in Supplementary Fig. 1. Induction ratios were calculated as described in the Methods. The *P* value of each field without an asterisk, $P > 0.05$ (one-way ANOVA with Dunnett's multiple comparisons post-test).

DDVA, regardless of the presence or absence of DDVA, indicating that the expression of *ddvT* is negatively regulated by DdvR.

The growth of Δ*ddvT* cells on lignin-derived aromatic compounds was examined. Growth retardation was observed, specifically with DDVA (Fig. 2d, Supplementary Fig. 5a). Furthermore, resting cells of Δ*ddvT* almost lost the ability to convert DDVA, whereas the same resting cells could convert other lignin-derived aromatic compounds as efficiently as the wild type (Fig. 2e, Supplementary Fig. 5b). Next, we evaluated DDVA uptake by Δ*ddvT* cells using a DDVA-uptake assay, which we had developed using the DdvR transcriptional regulation system with *lacZ* as the reporter[30]. When SYK-6 cells harbouring the reporter plasmid carrying *ddvR* and a transcriptional fusion of a *ligXa* promoter region and *lacZ* are incubated without DDVA, expression of *lacZ* is heavily repressed by DdvR. In the presence of DDVA, DDVA is incorporated into cells and the repression by DdvR is ended. Therefore, intercellular DDVA can be indirectly monitored by measuring LacZ activity. This assay showed that DDVA uptake by Δ*ddvT* cells was completely lost when they were incubated with 100 μM DDVA, as was seen with Δ*ddvK* cells (Fig. 2f). However, when Δ*ddvT* cells were incubated with 1 and 5 mM DDVA, the uptake of DDVA was estimated to be around 20% and 40% of that of the wild type, respectively. These results suggest that other transporter(s) are also involved in the outer membrane transport of DDVA under high concentrations. The introduction of *ddvT* into Δ*ddvT* cells restored their growth on DDVA, DDVA conversion and DDVA uptake (Fig. 2d, e, g). Additionally, the introduction of *ddvT* into SYK-6 cells considerably enhanced their DDVA conversion and uptake (Fig. 2e, g). Taken together, these results indicate that DdvT is involved in the uptake of DDVA. The growth of *ddvT*-complemented Δ*ddvT* on DDVA and its DDVA uptake (1 and 5 mM) were not fully recovered. These phenomena are most likely due to the low expression of *ddvT* resulting from the foreign promoter in the vector (Supplementary Figs. 6, 15b). We also assessed DDVA uptake by mutants of SLG_04460 and SLG_38050, whose transcription was induced 2.64- and 2.00-fold, respectively, during growth on DDVA, and by Δ*ompW*. However, these mutant cells exhibited almost the same level of DDVA uptake as that of the wild-type cells (Supplementary Figs. 2 and 7, Supplementary Table 2).

**Cellular localization of DdvT**. In order to investigate cellular localization of DdvT, a C-terminal His-tagged SLG_34540 encoding a TonB gene was first introduced into SYK-6 cells via a plasmid, so that the SLG_34540 product could act as an inner membrane marker. Using total and outer membrane fractions prepared from the above cells, we performed western blotting with anti-DdvT and anti-His6 antibodies. TonB was detected only in the total membrane fraction, whereas DdvT was detected in both fractions, indicating that DdvT is localized in the outer membrane (Fig. 2h, Supplementary Fig. 15c).

**The TonB box is important for DDVA uptake by DdvT**. At the N-terminus of SYK-6 TBDRs we found a conserved sequence consisting of XXXT (where X is a hydrophobic amino acid), which is similar to the TonB box[18]. DdvT contains IIVT spanning positions 40−43 (Supplementary Fig. 8). Because mutations of the TonB box have been reported to reduce substrate uptake and interactions between TBDR and TonB, alanine mutations were introduced into the highly conserved V42 and T43 residues of DdvT[31]. Both the V42A mutant and the T43A mutant showed reduced growth on DDVA and rates of DDVA conversion compared with Δ*ddvT* + *ddvT* (Fig. 3a, b, Supplementary Figs. 9, 15d). The T43A mutant also displayed a substantial decrease in DDVA uptake (Fig. 3c). When alanine mutations were introduced into both V42 and T43, the growth of this double mutant on DDVA, its DDVA conversion and its DDVA uptake decreased to levels comparable with Δ*ddvT*. Thus, V42 and T43 are important residues, which comprise the TonB box of DdvT. Based on all the results obtained above, we concluded that DdvT is the novel outer membrane transporter of DDVA.

In general, substrate uptake by porins is effective under high substrate concentrations because they transport substrates nonspecifically based on the substrate's concentration gradient. In contrast, TBDRs and substrate-specific channels are effective under low substrate concentrations since they show high substrate affinities of the order of nM and μM−mM, respectively[1,2,32]. The Δ*ddvT* mutant completely lost the ability to uptake and convert DDVA under low concentrations, indicating that SYK-6 depends on DdvT for the outer membrane transport of low concentrations of DDVA (Fig. 2e–g). Outer membrane transport systems are known to vary among bacteria. For example, substrate uptake by *Pseudomonas* is primarily dependent on substrate-specific channels, and therefore this genus exhibits high levels of antibiotic resistance by suppressing the influx of nonspecific substrates[33]. SYK-6 may utilize a number of TBDRs for its outer membrane substrate transport, instead of porins and substrate-specific channels. Based on the fact that the transcription of 16 TBDR genes other than *ddvT* was specifically induced during growth with lignin-derived aromatic compounds, it seems likely that these compounds are taken up by TBDRs (Fig. 1). It appears that SYK-6 switches TBDRs via substrate-specific inducible expression in order to facilitate the uptake of necessary nutrients. The genera *Alteromonas*, *Xanthomonas* and *Caulobacter* have large numbers of TBDR-like genes in their genomes, equal to or greater than the number present in SYK-6[9,10]. Presumably, the presence of various TBDRs with high substrate-affinity and -specificity is advantageous for bacteria, enabling them to survive in natural environments where nutrients are limited.

**Component genes of the TonB complex for DDVA uptake**. SYK-6 has six genes encoding putative TonBs (*tonB1−tonB6*), which transmit energy derived from the proton motive force to TBDRs (Supplementary Figs. 10 and 11). In addition, we found

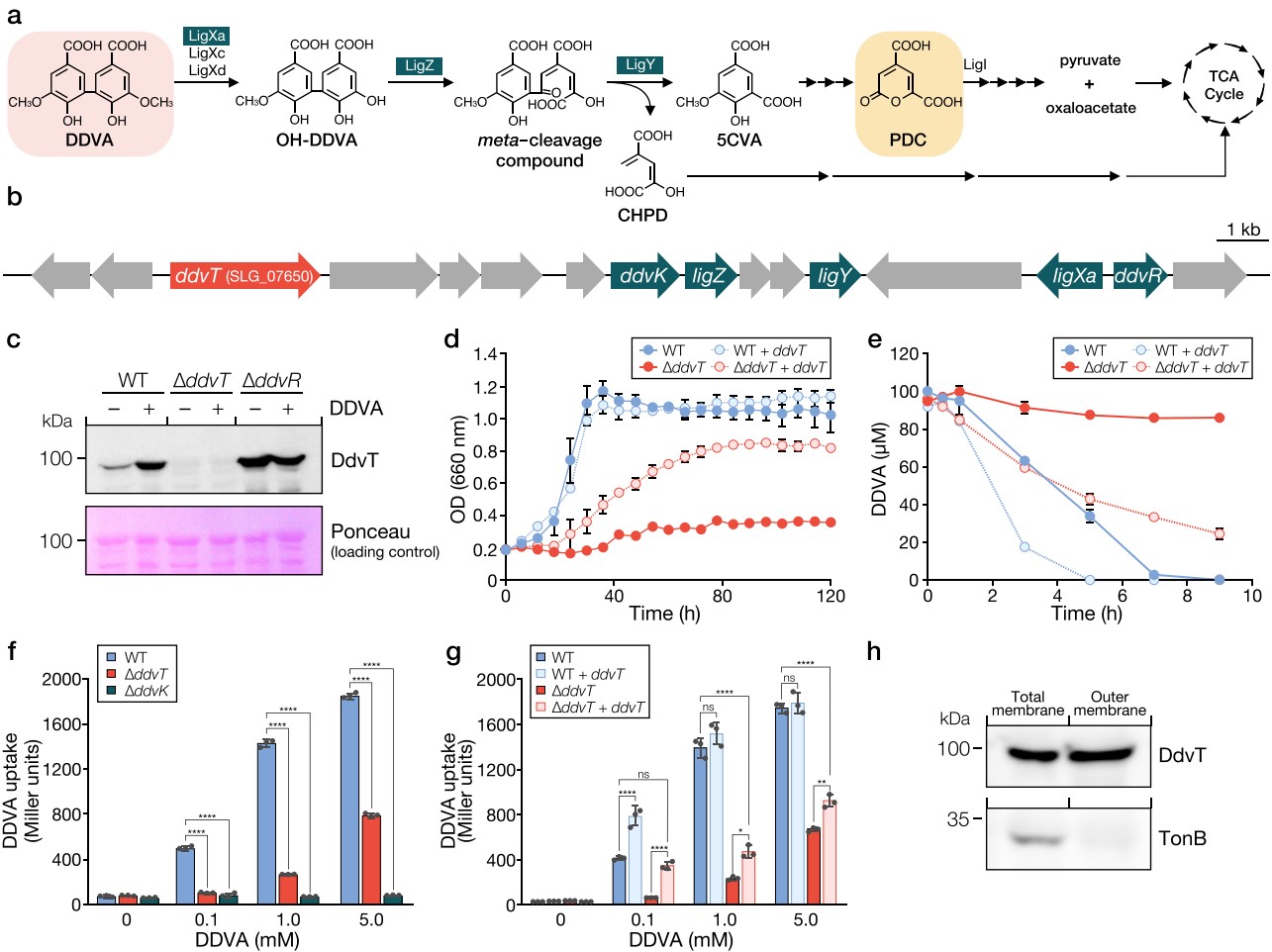

**Fig. 2** *ddvT* encodes the outer membrane transporter of DDVA. **a** The catabolic pathway of DDVA in SYK-6 cells. Enzymes: LigXa, oxygenase component of DDVA *O*-demethylase; LigXc, ferredoxin; LigXd, ferredoxin reductase; LigZ, OH-DDVA dioxygenase; LigY, *meta*-cleavage compound hydrolase; LigI, PDC hydrolase. Compounds: OH-DDVA, 2,2′,3-trihydroxy-3′-methoxy-5,5′-dicarboxybiphenyl; CHPD, 4-carboxy-2-hydroxypenta-2,4-dienoate; 5CVA, 5-carboxyvanillate; PDC, 2-pyrone-4,6-dicarboxylate. **b** Organization of the DDVA catabolic genes. Genes: SLG_07650 (*ddvT*), DDVA outer membrane transporter; *ddvK*, DDVA inner membrane transporter; *ddvR*, MarR-type transcriptional regulator. **c** Western blot analysis using anti-DdvT antibodies performed against total membrane fractions (10 μg of protein) obtained from cells of SYK-6, Δ*ddvT* and Δ*ddvR* incubated in LB supplemented with or without 1 mM DDVA. Ponceau S staining is shown as loading control. **d** Growth of Δ*ddvT* cells on DDVA. Cells of SYK-6(pSEVA338 [vector]), SYK-6(pS-ddvT), Δ*ddvT*(pSEVA338) and Δ*ddvT*(pS-ddvT) were cultured in Wx medium containing 5 mM DDVA. Cell growth was monitored by measuring the $OD_{660}$. **e** DDVA conversion by resting cells of Δ*ddvT*. Cells ($OD_{600} = 5.0$) of SYK-6(pSEVA338), SYK-6(pS-ddvT), Δ*ddvT*(pSEVA338) and Δ*ddvT*(pS-ddvT) were incubated with 100 μM DDVA. Portions of the reaction mixtures were collected and the amount of DDVA was measured by HPLC. **f** DDVA uptake by Δ*ddvT* cells. The β-galactosidase activities of cells of SYK-6(pS-XR), Δ*ddvT*(pS-XR) and Δ*ddvK*(pS-XR) incubated in Wx-SEMP with or without DDVA (0.1, 1.0 and 5.0 mM) were measured. **g** DDVA uptake by *ddvT*-complemented Δ*ddvT* cells. The β-galactosidase activities of cells of SYK-6 (pSEVA338 + pS-XR), SYK-6(pS-ddvT + pS-XR), Δ*ddvT*(pSEVA338 + pS-XR) and Δ*ddvT*(pS-ddvT + pS-XR) incubated in Wx-SEMP with or without DDVA (0.1, 1.0 and 5.0 mM) were measured. Each value is the average ± the standard deviation of $n = 3$ independent experiments. ns, $P > 0.05$, *$P < 0.05$, **$P < 0.01$, ****$P < 0.0001$ (one-way ANOVA with Dunnett's multiple comparisons post-test). **h** Cellular localization of DdvT in SYK-6 cells. Western blot analysis using anti-DdvT and anti-His6 antibodies was performed against the total membrane fraction (10 μg of protein) and the outer membrane fraction (10 μg of protein) obtained from SYK-6 cells harbouring pJB-tonB2His grown in LB containing 1 mM *m*-toluate.

two *exbB*-like genes (*exbB1* and *exbB2* [previously annotated as *tolQ*]) and three *exbD*-like genes (*exbD1*, *exbD2* and *exbD3* [previously annotated as *tolR*]). The gene products of *tonB*, *exbB* and *exbD* may constitute the TonB complex. Between them, *tonB1*, *exbB1*, *exbD1* and *exbD2* comprise an operon (Supplementary Fig. 12). We attempted to disrupt each of the *tonB* genes and succeeded in obtaining *tonB2−tonB6* mutants (Supplementary Fig. 2). Since a *tonB1* mutant was not obtained, *tonB1* may be essential for the growth of SYK-6. Among Δ*tonB2*−Δ*tonB6*, Δ*tonB2* exhibited reductions in growth on DDVA and in DDVA conversion (Fig. 4a, b). However, Δ*tonB2* also showed similar reductions in growth on other lignin-derived aromatic

compounds, lysogeny broth (LB) and SEMP (Supplementary Fig. 13). In the DDVA-uptake assay, Δ*tonB2* cells exhibited considerably higher LacZ activity than SYK-6 cells, implying the accumulation of DDVA in Δ*tonB2* cytoplasm (Fig. 4c). Because SLG_34550, just downstream of *tonB2*, has similarity with the *E. coli* TBDR gene (*fiu*) that encodes a siderophore transporter, *tonB2* appears to be involved in the uptake of iron (Supplementary Fig. 10). The disruption of *tonB2* seems to result in a reduction in iron uptake; this affects the activity of DDVA *O*-demethylase which contains a ferrous ion in its active centre[34].

Considering the possibility that DdvT interacts with multiple TonBs, we evaluated the growth of *tonB3-4-5-6* quadruple

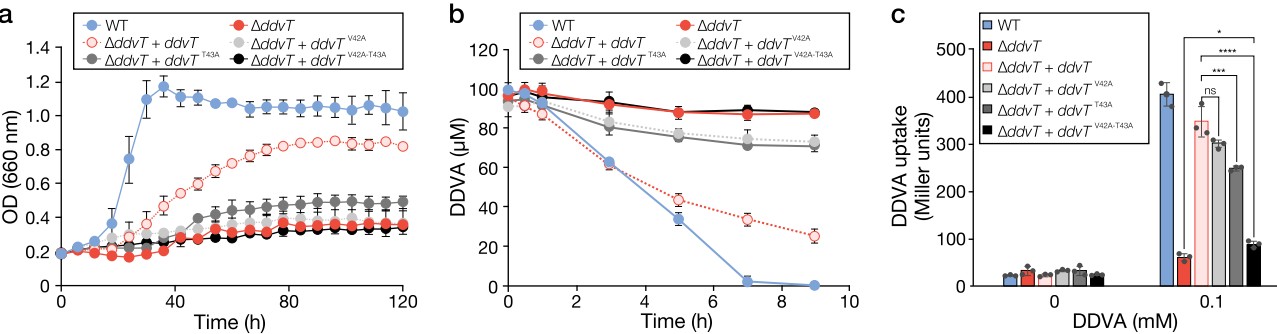

**Fig. 3** The TonB box in DdvT is important for DDVA uptake. **a** Growth of TonB box mutant cells on DDVA. Cells of SYK-6(pSEVA338 [vector]), Δ*ddvT* (pSEVA338), Δ*ddvT*(pS-ddvT), Δ*ddvT*(pS-ddvT$^{V42A}$), Δ*ddvT*(pS-ddvT$^{T43A}$) and Δ*ddvT*(pS-ddvT$^{V42A-T43A}$) were cultured in Wx medium containing 5 mM DDVA. **b** DDVA conversion by resting cells of TonB box mutants. Cells of SYK-6(pSEVA338), Δ*ddvT*(pSEVA338), Δ*ddvT*(pS-ddvT), Δ*ddvT*(pS-ddvT$^{V42A}$), Δ*ddvT*(pS-ddvT$^{T43A}$) and Δ*ddvT*(pS-ddvT$^{V42A-T43A}$) were incubated with 100 μM DDVA. **c** DDVA uptake by TonB box mutant cells. The β-galactosidase activities of cells of SYK-6(pSEVA338 + pS-XR), Δ*ddvT*(pSEVA338 + pS-XR), Δ*ddvT*(pS-ddvT + pS-XR), Δ*ddvT*(pS-ddvT$^{V42A}$ + pS-XR), Δ*ddvT*(pS-ddvT$^{T43A}$ + pS-XR) and Δ*ddvT*(pS-ddvT$^{V42A-T43A}$ + pS-XR) incubated in Wx-SEMP with or without 100 μM DDVA were measured. Each value is the average ± the standard deviation of $n = 3$ independent experiments. ns, $P > 0.05$, *$P < 0.05$, ***$P < 0.001$, ****$P < 0.0001$ (one-way ANOVA with Dunnett's multiple comparisons post-test).

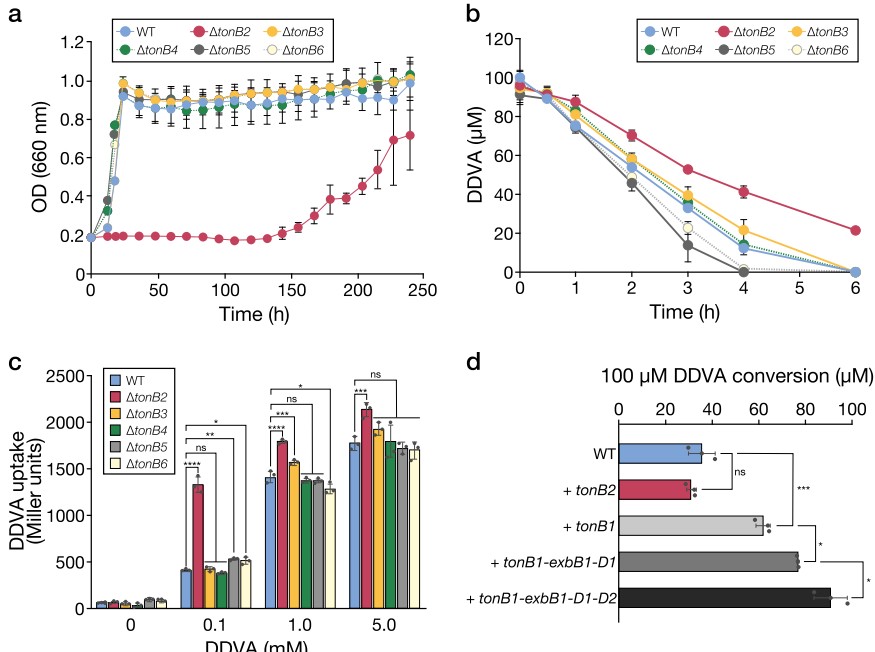

**Fig. 4** Identification of components of the TonB complex necessary for DDVA uptake. **a** Growth of five *tonB* mutants on DDVA. Cells of SYK-6 and the *tonB* mutants were cultured in Wx medium containing 5 mM DDVA. Cell growth was monitored by measuring the OD$_{660}$. **b** DDVA conversion by resting cells of *tonB* mutants. Cells of SYK-6 and the *tonB* mutants were incubated with 100 μM DDVA. Portions of the reaction mixtures were collected and the amount of DDVA was measured by HPLC. **c** DDVA uptake by *tonB* mutant cells. The β-galactosidase activities of cells of SYK-6(pS-XR) and the *tonB* mutants(pS-XR) incubated in Wx-SEMP with or without DDVA (0.1, 1.0 and 5.0 mM) were measured. **d** DDVA conversion by SYK-6 resting cells harbouring pJB-tonB1 carrying *tonB1*, pJB-tonB2 carrying *tonB2*, pJB-t1-D1 carrying *tonB1*, *exbB1* and *exbD1* or pJB-t1-D12 carrying *tonB1*, *exbB1*, *exbD1* and *exbD2*. The concentrations of DDVA converted after 5 h are shown. Each value is the average ± the standard deviation of $n = 3$ independent experiments. ns, $P > 0.05$, *$P < 0.05$, **$P < 0.01$, ***$P < 0.001$, ****$P < 0.0001$ (one-way ANOVA with Dunnett's multiple comparisons post-test).

mutant (Δ*tonB3456*) cells on DDVA and their DDVA uptake (Supplementary Fig. 14a, b). However, Δ*tonB3456* grew as well as the wild type, with DDVA uptake equivalent to the level seen in the wild type. In addition, the growth of *tonB2-3-4-5-6* quintuple mutant cells on DDVA and their DDVA uptake were comparable to those of Δ*tonB2* cells (Supplementary Fig. 14c, d). These results suggest that *tonB1* plays a major role in the outer membrane transport of DDVA. To clarify the involvement of TonB1 in the uptake of DDVA, we evaluated the ability of resting SYK-6 cells harbouring a plasmid carrying *tonB1* to

convert DDVA. The amount of DDVA converted in *tonB1*-overexpressing cells increased ca. 1.8-fold (61 ± 3.2 μM) compared with the amount converted by wild-type cells (35 ± 5.7 μM) after 5 h (Fig. 4d). Furthermore, the amount of DDVA converted by cells overexpressing *tonB1*−*exbB1*−*exbD1* and *tonB1*−*exbB1*−*exbD1*−*exbD2* was ca. 1.2- (77 ± 0.4 μM) and 1.4-fold (91 ± 7.1 μM) higher, respectively, than that converted by cells overexpressing *tonB1* only. On the other hand, the overexpression of *tonB2* had no effect on DDVA conversion (30 ± 2.1 μM). These results suggest that TonB1, and possibly

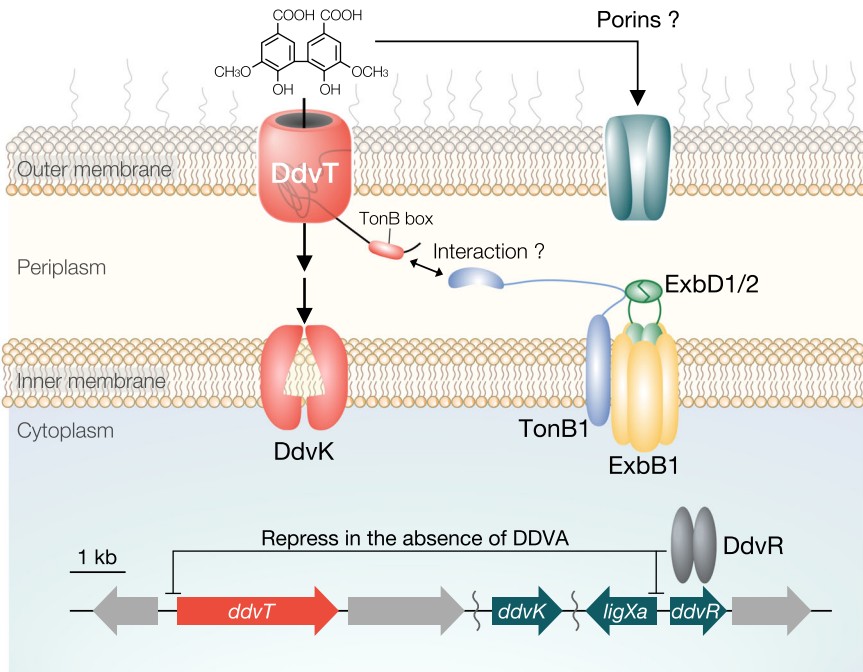

**Fig. 5** The proposed DDVA uptake pathway in SYK-6 cells. Outer membrane transport of DDVA is mediated by DdvT; the energy necessary for the transport derived from the proton motive force appears to be transmitted from the inner membrane-localized TonB complex consisting of TonB1−ExbB1−ExbD1/ExbD2. In the inner membrane, DdvK takes up DDVA into the cytoplasm. In the presence of high concentrations of DDVA (in the order of > μM), unidentified outer membrane transporters such as porins are likely be involved in DDVA uptake. DdvR represses the expression of both *ddvT* and *ligXa* in the absence of DDVA. A schematic diagram of the TonB complex was drawn based on the papers by Celia et al.[3,53].

ExbB1, ExbD1 and ExbD2 are involved in the uptake of DDVA (Fig. 5).

**Overexpression of *ddvT* enhances production of PDC from DDVA.** A enhancement in the conversion and uptake of DDVA was observed following the overexpression of *ddvT* in SYK-6 cells (Fig. 2e, g). Based on this result, we evaluated the effect of *ddvT* overexpression on the production of PDC, which is a promising platform chemical produced from lignin, using an SYK-6 mutant of the PDC hydrolase gene (ΔligI), which accumulates PDC. When ΔligI cells harbouring a plasmid carrying *ddvT* were grown in Wx-SEMP medium in the presence of 1 mM DDVA, the amount of DDVA converted and PDC produced by these cells after 20 h increased ca. 1.3-fold compared with ΔligI cells (Fig. 6a). The cell yield also increased ca. 1.3-fold, suggesting that the cells efficiently utilized 4-carboxy-2-hydroxypenta-2,4-dienoate (CHPD) generated from the *meta*-cleavage compound of DDVA in addition to SEMP (Figs. 2a, 6b).

To date, TBDRs have mainly been investigated as a drug discovery target in the fight against pathogens because they function in the initial step of acquiring iron and other nutrients essential for growth. In contrast, to date no attempt has been made to utilize TBDRs for improving microbial conversion. Our results clearly demonstrate, however, that the overexpression of a TBDR gene encoding an outer membrane transporter is an effective means of improving the rate of metabolite production from lignin-derived aromatic compounds. Recently, there have been some reports of observations or at least suggestions of the possibility that TBDRs are involved in the uptake of chitin, cellulose oligomers and dioxin in non-pathogenic Gram-negative bacteria[16,35,36]. Our findings will contribute not only to the understanding of the diverse functions of TBDRs but also to the application of TBDRs for improvements in the efficiencies of microbial production of value-added products and bioremediation.

## Methods

**Bacterial strains, plasmids and culture conditions**. The strains and plasmids used in this study are listed in Supplementary Table 3 and the PCR primers are listed in Supplementary Table 4. *Sphingobium* sp. SYK-6 and its mutants were grown at 30 °C with shaking (160 rpm) in LB or Wx minimal medium containing SEMP[37]. Media for transformants of SYK-6 and its mutants was supplemented with 50 mg l$^{-1}$ kanamycin (Km) or 30 mg l$^{-1}$ chloramphenicol (Cm). *E. coli* strains were cultured in LB at 37 °C. Media for *E. coli* transformants carrying antibiotic resistance markers was supplemented with 25 mg l$^{-1}$ Km or 30 mg l$^{-1}$ Cm. DDVA, PR, DCA and HMPPD were chemically synthesized from ethyl vanillate, coniferyl aldehyde, coniferyl aldehyde and homovanillic acid, respectively[38–41]. PDC was obtained from PCA by incubating with *P. putida* PpY1100 cells harbouring a plasmid carrying the protocatechuate 4,5-dioxygenase gene (*ligAB*) and the 4-carboxy-2-hydroxymuconate-6-semialdehyde dehydrogenase gene (*ligC*)[26]. Other aromatic compounds were purchased from the Tokyo Chemical Co., Ltd. or FUJIFILM Wako Pure Chemical Corporation. *Sphingobium* sp. SYK-6 is available from Biological Resource Center at National Institute of Technology and Evaluation and RIKEN BioResource Center under strain numbers NBRC 103272 and JCM 17495, respectively.

**Construction of mutants**. To construct plasmids for gene disruption, ca. 1-kb fragments carrying upstream and downstream regions of each gene were amplified by PCR using SYK-6 genome DNA as a template and the primer pairs shown in Supplementary Table 4. The resulting fragments were inserted into the *Bam*HI site in pAK405 by In-Fusion cloning (TaKaRa Bio, Inc.). These plasmids were independently introduced into SYK-6 cells and its mutants by triparental mating, and transformants generated by the first homologous recombination were selected on an LB agar medium containing Km and 12.5 mg l$^{-1}$ nalidixic acid. Selection of candidate mutants was performed according to the reported method with slight modifications described below[42]. A mixture of transformants was cultured in LB liquid medium containing 100 mg l$^{-1}$ streptomycin, and then plated on an LB agar medium containing 100 mg l$^{-1}$ streptomycin to select for the second homologous recombination event. Resulting colonies were streaked on both an LB agar medium containing streptomycin and an LB agar medium containing Km, and Km-sensitive colonies were analysed by colony PCR using the primer pairs shown in Supplementary Table 4. The plasmids for gene complementation of Δ*ddvT* and Δ*tonB2* (Supplementary Table 3) were introduced into mutants by electroporation.

**Sequence analysis and construction of the phylogenetic tree**. Sequence analysis was performed using the MacVector program version 15.5.2 (MacVector, Inc.). Sequence similarity searches, pairwise alignments and multiple alignments were performed using the BLAST program[43], the EMBOSS program[44] and the Clustal

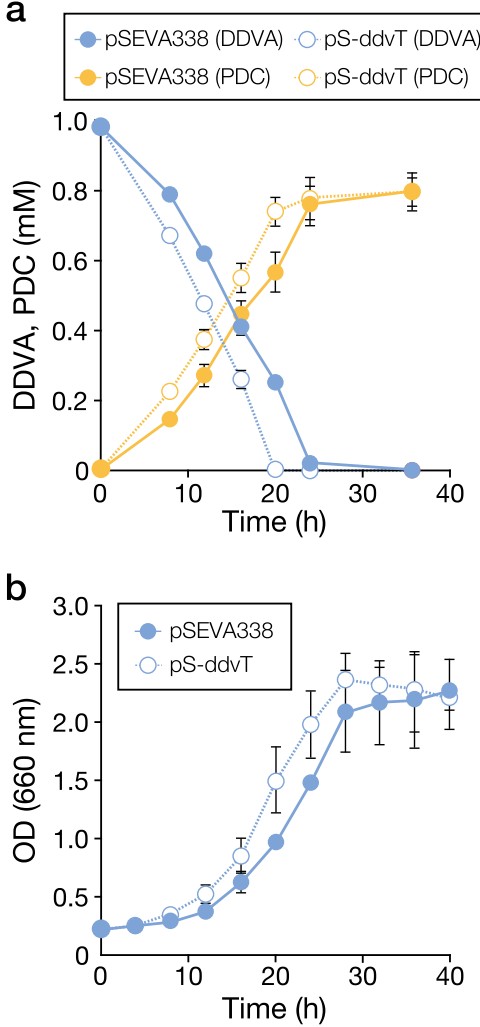

**Fig. 6** The effect of *ddvT* overexpression on PDC production. **a** Cells of Δ*ligI* (pSEVA338 [vector]) and Δ*ligI*(pS-ddvT) were cultured in Wx-SEMP containing 1 mM DDVA and 0.5 mM *m*-toluate. The amount of DDVA converted and PDC produced in the cultures was measured using HPLC. **b** Growth of Δ*ligI*(pSEVA338) and Δ*ligI*(pS-ddvT) in the above-mentioned medium. Each value is the average ± the standard deviation of $n = 3$ independent experiments.

Omega program[45], respectively. For phylogenetic analysis of TBDRs, multiple alignments were performed using the Clustal W program in MEGA X[46], and then a phylogenetic tree was generated using the neighbour-joining algorithm of MEGA X, employing 1000 bootstrap replicates. Putative transmembrane segments were predicted using the TMHMM program[47] and the BOCTOPUS2 program[48], and signal sequence prediction was performed using SignalP-5.0[49].

**DNA microarray analysis.** SYK-6 cells were grown in Wx-SEMP at 30 °C until the optical density of the culture reached 0.5 at 600 nm ($OD_{600}$). Cultures were further incubated with 5 mM DDVA, HMPPD, GGE, FA, AV or VN for 6 h; 5 mM PCA for 2 h; 2 mM DCA or PR for 2 h; or without substrates for 2 h. Total RNA was isolated from the resulting cells from three independent cultures and used for DNA microarray analysis after DNase I treatment[50]. Aminoallyl-labelled cDNA was synthesized by reverse transcription using total RNA (6 μg), mixture of two kinds of random hexamers, normal GC content (Invitrogen) and high GC content (70%) (Sigma), 5-(3-aminoallyl)-dUTP (Ambion) and PrimeScript II reverse transcriptase (TaKaRa Bio Inc.). The RNA template was then degraded through incubation with 0.2 N NaOH and 0.1 M EDTA, followed by a neutralization using 1 M HEPES (pH 7.5). For a control of microarray hybridization, 4 μg of fragmented genomic DNA of SYK-6 was labelled using 5-(3-aminoallyl)-dUTP and Klenow fragment. Cy3 and Cy5 dyes were coupled to the aminoallyl-dUTP in the cDNA and genomic DNA, respectively, in the presence of 0.1 M sodium bicarbonate (pH 9.0). The unlabelled dyes were removed using the QIAquick PCR purification system (Qiagen). Hybridizations were performed in a GeneTac HybStation instrument

(Genomic Solutions). Hybridized arrays were scanned using a GenePix 4000B scanner (Axon Instruments), and the spot intensities were quantified using Imagene 6.1 (BioDiscovery). The expression patterns in cells grown in Wx-SEMP plus each lignin-derived aromatic compound were compared with those of cells grown in Wx-SEMP using an in silico analysis performed with the linear model for microarray analysis loess (subgrid) method using ArrayPipe 2.0[51]. Average normalized expression ratios (treatment/control) were calculated for each gene and tested for any significant variation between treatments (one-way ANOVA with Dunnett's multiple comparisons post-test). Each value was obtained from $n = 3$ independent experiments.

**RT-PCR analysis.** Total RNA was isolated from SYK-6 cells grown in Wx-SEMP for 8 h using an Illumina RNAspin Mini RNA isolation kit (GE Healthcare). To remove any contaminating genomic DNA, the samples were treated with DNase I (TaKaRa Bio, Inc.). Total RNA (4 μg) was reverse transcribed using SuperScript IV reverse transcriptase (Invitrogen) with random hexamer primers. The cDNA was purified using a NucleoSpin Gel and PCR Clean-up kit (Takara Bio, Inc.). PCR was performed with the cDNA, specific primers (Supplementary Table 4) and Gflex DNA polymerase (Takara Bio, Inc.). The resulting DNA was subjected to 0.8% agarose gel electrophoresis.

**Growth measurement.** The cells of SYK-6, its mutants and complemented strains were grown in LB for 24 h, harvested by centrifugation at $4800 \times g$ for 5 min, washed twice with Wx medium and resuspended in 3 ml of the same medium. The cells were then inoculated in Wx medium containing 5 mM DDVA, SA, SN, VA, VN, PCA or FA to an $OD_{660}$ of 0.2. SYK-6 exhibits auxotrophy for methionine when grown in a methoxy-group-free substrate, so 20 mg l$^{-1}$ methionine was added to the medium for growth on PCA. Cells were incubated at 30 °C with shaking (60 rpm) and cell growth was monitored every hour by measuring the $OD_{660}$ with a TVS062CA biophotorecorder (Advantec Co., Ltd.). For the analysis of complemented strains of Δ*ddvT*, cells were grown in Wx medium containing Cm and 0.5 mM *m*-toluate (an inducer of the $P_m$ promoter in pSEVA338).

**Resting cell assay.** The cells of SYK-6 and its mutants were grown in LB for 20 h, harvested by centrifugation at $4800 \times g$ for 5 min, washed twice with 50 mM Tris-HCl buffer (pH 7.5) and resuspended in 1 ml of the same buffer. The cells were then inoculated in 50 mM Tris-HCl buffer (pH 7.5) containing 100 μM DDVA to an $OD_{600}$ of 5.0 and incubated for 6 h. For conversion of 200 μM DCA, 200 μM GGE and 100 μM HMPPD, cells were inoculated to an $OD_{600}$ of 0.5, 2.0 and 2.0, respectively, and then the mixtures were incubated for 3, 6 and 3 h, respectively. Samples were collected periodically and the reactions were stopped by centrifugation at $18,800 \times g$ for 10 min. The supernatants were diluted fivefold in water, filtered, and analysed by high-performance liquid chromatography (HPLC). For the analysis of the complemented strains of Δ*ddvT* and SYK-6 harbouring a plasmid carrying *tonB* or component genes of the TonB complex, the cells grown in LB containing Km or Cm and 0.5 mM *m*-toluate were employed.

**HPLC conditions.** HPLC analysis was performed using an Acquity UPLC system (Waters Corporation) with a TSKgel ODS-140HTP column (2.1 by 100 mm; Tosoh Corporation). All analyses were carried out at a flow rate of 0.5 ml min$^{-1}$ except the analysis of PDC (0.3 ml min$^{-1}$). The mobile phase was a mixture of solution A (acetonitrile containing 0.1% formic acid) and solution B (water containing 0.1% formic acid) under the following conditions. For the analysis of conversion of DDVA, 0–2.5 min, 15% A. For the analysis of conversion of GGE, 0–3.2 min, linear gradient from 5 to 40% A; 3.2–6.0 min, decreasing gradient from 40 to 5% A; 6.0–7.0 min, 5% A. For the analysis of conversion of DCA, 0–3.0 min, 25% A. For the analysis of conversion of HMPPD, 0–2.5 min, 10% A. For the analysis of PDC accumulation, the mobile phase was a mixture of water (85%) and acetonitrile (15%) containing 0.1% phosphoric acid. DDVA, GGE, DCA, HMPPD and PDC were detected at 265, 279, 280, 279 and 315 nm, respectively.

**DDVA-uptake assay.** Cells of SYK-6 and its mutants harbouring pS-XR grown in LB containing Km for 20 h were harvested by centrifugation at $4800 \times g$ for 5 min, washed twice with Wx medium and resuspended in 1.0 ml Wx-SEMP. The cells were then inoculated in Wx-SEMP with or without DDVA (0.1, 1.0 or 5.0 mM) to an $OD_{600}$ of 2.0. Samples were incubated at 30 °C with shaking (1500 rpm) for 3 h. The β-galactosidase activity of the cells was measured using 2-nitrophenyl-β-D-galactopyranoside as the substrate, according to a modified Miller assay (https://openwetware.org/wiki/Beta-Galactosidase_Assay_(A_better_Miller))[30]. β-galactosidase activity is expressed as Miller units. For complementation analysis, cells were grown in LB containing Km, Cm and 0.5 mM *m*-toluate, and used for the assay.

**Alanine mutagenesis.** Alanine mutagenesis for the residues in the predicted TonB box was performed by inverse PCR using pS-ddvT as a template and the mutation primers listed in Supplementary Table 4. The plasmids carrying mutated *ddvT* were

introduced into Δ*ddvT* harbouring pS-XR by electroporation. The growth of these strains on DDVA and their ability to uptake and convert DDVA were evaluated.

**Western blot analysis**. A peptide corresponding to residues 67–84 (AER-GATNIGDFLNEVPSF) of DdvT was synthesized and used as an antigen to obtain antisera against DdvT in rabbits (Cosmo Bio, Inc.). Anti-DdvT-peptide antibodies were obtained by purification of the antiserum using peptide affinity column chromatography (Cosmo Bio, Inc.). Western blot analysis using anti-DdvT antibodies was performed against total membrane fractions prepared from SYK-6, Δ*ddvT* and Δ*ddvR* cultured in the presence or absence of DDVA. The cells were grown in LB and 1 mM DDVA was added when the $OD_{600}$ of the culture reached 0.5; they were then incubated for a further 12 h. Cells were harvested by centrifugation, washed with 50 mM Tris-HCl buffer (pH 7.5) and resuspended in the same buffer. The cells were disrupted by sonication and cell lysate was obtained. After the cell lysate was centrifuged at $18,800 \times g$ for 10 min, the resulting supernatant was ultracentrifuged at $120,000 \times g$ for 60 min to obtain the total membrane fraction. Proteins were separated by SDS-PAGE and transferred onto a PVDF membrane (Bio-Rad Laboratories) by electroblotting. The proteins on the membrane were stained with Ponceau S. After destaining with 0.1 M NaOH and washing with water, the PVDF membrane was blocked by incubation with TBS buffer (pH 7.4) containing 0.1% Tween20 (TBST buffer) and 5% blocking agent (GE Healthcare) for 1 h at room temperature, and then incubated with the same buffer containing primary anti-DdvT antibodies ($0.25 \,\mu g \, ml^{-1}$). After incubation for 1 h, the membrane was washed four times with TBST buffer, and then incubated with the same buffer containing horseradish peroxidase-conjugated goat anti-rabbit IgG secondary antibodies (Invitrogen, $0.2 \,\mu g \, ml^{-1}$) for 1 h. Following incubation, the membrane was washed four times with TBST buffer, and horseradish peroxidase activity was detected by chemiluminescence using the ECL Western Blotting Detection System (GE Healthcare) with a LumiVision PRO image analyser (Aisin Seiki Co., Ltd). For detection of DdvT in the complemented strains and the TonB box mutants, the cells grown in LB containing Cm and 0.5 mM *m*-toluate were used for the assay. Protein concentrations were determined by the Bradford method using a Bio-Rad protein assay kit or Lowry's assay with a DC protein assay kit (Bio-Rad Laboratories).

**Cellular localization of DdvT in SYK-6**. SYK-6 cells harbouring pJB-tonB2His that carried *tonB2* fused with a His6 tag at its C-terminus in pJB861 were grown in LB containing Km and 1.0 mM *m*-toluate to an $OD_{600}$ of 0.5. The outer membrane fraction was prepared according to the method reported by Hashimoto et al.[52]. The cells were harvested by centrifugation at $4800 \times g$ for 10 min, washed twice with water and incubated with 26 mM Tris-HCl buffer (pH 8.3) containing 438 mM sucrose, 1.5 mM EDTA and $0.22 \, mg \, ml^{-1}$ lysozyme at 30 °C for 1 h. The resulting solution was centrifuged at $19,000 \times g$ for 60 min to remove spheroplasts, and then the supernatant was ultracentrifuged at $120,000 \times g$ for 60 min to obtain the outer membrane fraction. The total membrane fraction was prepared as described above. Western blot analysis was performed against the prepared outer membrane fraction and total membrane fraction using anti-DdvT and anti-His6 antibodies (Invitrogen, $1.0 \,\mu g \, ml^{-1}$) as primary antibodies. Horseradish peroxidase-conjugated goat anti-mouse IgG antibodies (Invitrogen, $0.04 \,\mu g \, ml^{-1}$) were used as the secondary antibodies for the anti-His6 antibodies.

**PDC production**. Δ*ligI* cells harbouring pSEVA338 or pS-ddvT were grown in LB containing Cm for 24 h. The cells were harvested by centrifugation at $4800 \times g$ for 5 min, washed twice with Wx medium and resuspended in 3 ml of the same medium. The cells were inoculated to an $OD_{660}$ of 0.2 in 5 ml Wx medium containing SEMP, 1 mM DDVA, Cm and 0.5 mM *m*-toluate and incubated at 30 °C with shaking (60 rpm). Cell growth was periodically monitored by measuring the $OD_{660}$. Samples were periodically collected by centrifugation, diluted, filtered and analysed by HPLC.

**Statistics and reproducibility**. All our results were obtained from $n = 3$ independent experiments. Statistic tests were performed with Graphpad Prism8 (Graphpad software). One-way ANOVA with Dunnett's multiple comparisons post-test was used as shown in figure legends. $P < 0.05$ was considered statistically significant.

**Reporting summary**. Further information on research design is available in the Nature Research Reporting Summary linked to this article.

## Data availability
The DNA microarray data have been registered in the Gene Expression Omnibus under accession number GSE134094. All data supporting this study are available within the article and its Supplementary Information or are available from the corresponding author upon request.

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

## Acknowledgements

We thank Koki Shibata for assistance with the construction of the *ompW* mutant. We also thank Kenta Tanatani for isolating total RNA to perform the RT-PCR analysis. This work was supported by JSPS KAKENHI Grant Numbers 15H04473, 19H02867 and 19J11312.

## Author contributions

E.M. conceived and supervised the study. M.F., N.K. and E.M. designed the study, performed data analysis and wrote the manuscript. M.F. performed the experiments, with the following exceptions. K.M. constructed the *ddvT* mutant. N.K. and H.H. performed the DNA microarray analysis. S.H. synthesized the lignin-derived aromatic compounds used in this study. K.M. helped to interpret the data and discussed the results. All authors reviewed the paper.

## Competing interests

The authors declare no competing interests.
