## [Peer Review File · Communications Biology]

Reviewers' comments:

Reviewer #1 (Remarks to the Author):

Review of Fujita et al., A novel outer membrane transport system for an aromatic compound mediated by a TonB-dependent receptor

Summary:

Fujita et al. present a very thorough and systematic approach to the characterization of 5,5'-dehydrodivanillate (DDVA) uptake by DdvT from *Sphingobium* sp. SYK-6. A genome-wide phylogenetic assessment combined with DNA microarray analysis was leveraged to identify putative TonB-dependent receptors (TBDRs), 17 of which were transcriptionally induced in the presence of DDVA. One of these targets, DdvT, is shown to be an outer membrane TBDR via assays for growth on DDVA by DdvT mutant cells, DDVA uptake rates as measured by a lacZ reporter system, and OM localization as measured by Anti-DdvT Western Blots. The authors then demonstrate that two hydrophobic residues in the N-terminus TonB box of DdvT and TonB1 are important for DdvT-mediated DDVA uptake. Given this, the authors present a model for the uptake of DDVA and leverage this knowledge toward faster consumption and subsequent accumulation of PDC from DDVA. The manuscript is well written and edited. Only minor concerns and corrections are requested.

Minor concerns and edits:

1. Uptake of DDVA is presented only as Miller units using the lacZ reporter assay. While this is a clever solution for measuring intracellular DDVA concentration, how does a Miller unit correlate to μM of DDVA in the cytoplasm?
2. How many biological replicates were analyzed for the DNA microarray analyses? Please specify in the methods (p. 19) and Fig. 1 caption (p. 32).
3. Based on data presented in Fig. 2f, the authors hypothesize that there may be unidentified OM porins responsible for DDVA uptake at high concentrations. Is it also possible that passive diffusion of DDVA across the outer membrane could occur at 1 mM and 5 mM concentrations?
4. On lines 190-192, the authors state that the combination of V42A and T43A mutations in the TonB box of DdvT decrease DDVA uptake "to levels comparable with ΔddvT ". However, in Fig. 3c no tests are presented for uptake rates of the TonB point mutations against ΔddvT . Instead, all comparisons are made against $\Delta\text{ddcT}+\text{ddvT}$, which is not the comparison discussed in the text. Either or both comparison(s) could be justified, but please be consistent in the text and figure for the presented comparison(s).
5. The conclusion is presented that over-expression of DdvT is "is an effective means for improving the efficiency of metabolite production..." (lines 261-267) and contributes to "improved efficiency of platform chemical production" (abstract, line 33). However, only fold-changes as compared to an empty vector control are presented for yield (1.3-fold increase), and "amount converted" (also 1.3-fold yield). It is difficult to understand how these values were derived from examining Fig. 6 alone, where it appears all of the DDVA was converted in both the empty-vector and DdvT over-expression strain. It seems the rate of PDC production may be the most improved metric, but this is not discussed. To support your claims, please provide a corresponding T/R/Y value in addition to the fold-changes presented in the text.

6. Minor edits:

- a. Please define or provide a reference for "SEMP" in the methods (line 286) and Fig. 1 legend.
- b. Please define the test used to compute p-values presented in Fig. 1 (lines 598-599).
- c. In Fig. 2, what do the error bars represent? Please include in figure legend (p. 34-35).

Reviewer #2 (Remarks to the Author):

In this paper the authors show that the outer membrane transport of lignin-derived DDVA in *Sphingobium* sp. SYK-6 occurs via the putative TonB dependent transporter: DdvT.

This was confirmed by first eliminating the possibility that the transporter ompW was involved in DDVA transport. SYK-6 has ompW-like genes that exhibit similarities of aromatic-compound porins. An ompW mutant was produced and the effect on SYK-6 growth in the presence of ligand-derived aromatic compounds including DDVA was monitored. Cell growth in the mutant was similar to that of WT. Moreover, disruption of *ddvT* confirmed DdvT's role in DDVA transport. In the absence of *ddvT* resting SYK-6 cells lost the ability to convert DDVA. In addition, DDVA uptake was significantly reduced. These results showed that unlike the ompW deletion, the deletion of *ddvT* was essential to DDVA uptake and cell growth.

In addition, to confirm DdvT as a TBDT the authors went about mutating residues in the TonB box at the N-term of the protein. It is well established that the TonB box is essential in transport involving TBDTs. The alanine mutations introduced disrupted DdvT's ability for DDVA uptake. These findings do support that DdvT may in fact be a TBDT. Mutagenesis studies of the putative TonB complex components also confirmed their role in DDVA uptake.

Lastly the authors show that overexpression of *ddvT* increases uptake and conversion of DDVA. This leads to an overexpression of PDC, a platform chemical produced from lignin.

Overall, the authors show that DdvT is a novel TBDT involved in the transport of lignin derived DDVA. However, TBDTs have been extensively studied and numerous crystal structures of various TBDTs in complex with their ligands have been published over the years. Though the findings of these studies are interesting, the authors have not shown anything novel about DdvT as a new TBDT. This manuscript would be better suited to a journal such as *Journal of Bacteriology*, where a large number of similar papers are published.

Specific comments

[1] In the abstract the authors claim uptake of aromatic compounds in TBDTs was previously unknown; however, this 2010 paper suggests the role of TBDRs in *Sphingomonas wittichii* RW1, specifically as transporters of aromatic hydrocarbons and/or their breakdown products.

Genome Sequence of the Dioxin-Mineralizing Bacterium *Sphingomonas wittichii* RW1
Todd R. Miller, Arthur L. Delcher, Steven L. Salzberg, Elizabeth Saunders, John C. Detter, Rolf U. Halden
DOI: 10.1128/JB.01030-10

The authors later speak of upregulation of TBDR-like genes in *S. wittichii* RW1 and *Pseudomonas*

strains, but the way the manuscript is presented, they are overstating the novelty of their findings.

[2] There is no clear explanation of the potential benefits of developing this system as a platform for lignin-derived compounds.

[3] Parts of the manuscript are overly speculative:

Page 12, lines 208-210: just because TBDR genes are upregulated by growth on lignin-derived compounds does not mean that those genes take up these compounds. This reviewer has unpublished data on another system refuting such a claim.

Pages 13-15: the whole section on component genes of the TonB complex necessary for DDVA uptake is poorly executed and overly speculative.

[4] The manuscript would be strengthened by routine assays in the TBDR field: purify the protein, perform binding studies, etc.

[5] Figure 4: Although not discussed at all in the manuscript (so why is this included as a figure?), the TonB5 mutant appears to convert DDVA faster than wildtype. What is going on here?

[6] Figure 5: The schematic of the DDVA uptake pathway shown in Figure 5 does not accurately depict the proposed model of the TonB complex in the inner membrane of Gram-negative bacteria. The authors should reference the following article: Nature volume 538, pages 60–65 (06 October 2016) for a better understanding of the components of the TonB system. This may help in improving their schematic.

[7] Finally, the inclusion of 14 supplementary figures and 4 tables is excessive and suggests that this manuscript is derived from a student's PhD thesis. Among the unnecessary supplementary figures are table 4, figure 2, and figure 4.

Reviewer #3 (Remarks to the Author):

The work from Fujita et al. reports on the identification of a transporter (DdvT) involved in the import of a biphenyl aromatic intermediate in the biodegradation of lignin (5,5'-dehydrodivanillate, DDVA) in *Sphingobium* SYK-6. This transporter is located in the outer membrane and belongs to a TonB-dependent transport system, demonstrating a new role for this type of transporters. Moreover, genes encoding the components of the TonB complex located in the inner membrane and involved in the transmission of energy to DdvT were identified. The authors increased the production efficiency of the valuable platform chemical 2-pyrone-4,6-dicarboxylate from DDVA by overexpressing DdvT in a mutant strain of SYK-6.

This is a very nice work which provides important information on the transport mechanisms involved in the uptake of aromatic compounds derived from lignin in SYK-6. Claims are novel and will be of interest to others in the community and the wider field, including lignin valorization. Several uncharacterized TonB-dependent transporters have been identified, and the authors have nicely characterized in detail the receptor and its TonB complex responsible for the uptake of DDVA. Experiments were nicely designed and the data support the conclusions.

The manuscript is overall suitable for publication in *Communications Biology*. I would however

encourage the authors to include a loading control for western blots shown in figures 2c, S6 and S9.

Response to reviewer's comments

We appreciate the helpful and vital suggestions from the reviewers for our manuscript COMMSBIO-19-1060-T. The following are our responses to the reviewer's comments, and we have modified our manuscript according to the comments. In the revised manuscript file with the changes marked, the red letters indicate portions that correspond to the reviewer's comments or corrected by us for improvement.

To Reviewer #1

Q1) Uptake of DDVA is presented only as Miller units using the lacZ reporter assay. While this is a clever solution for measuring intracellular DDVA concentration, how does a Miller unit correlate to μM of DDVA in the cytoplasm?

We have confirmed that the LacZ activities of SYK-6 harboring pS-XR correlated to the concentration of DDVA outside of the cells. However, it is technically challenging to quantify the DDVA concentration in the cytoplasm. We think that this method can be applied to determine the relative uptake-ability of the cells.

Q2) How many biological replicates were analyzed for the DNA microarray analyses? Please specify in the methods (p. 19) and Fig. 1 caption (p. 32).

L323, L332, L625: The DNA microarray analysis was performed in triplicate. We have added this to the Methods section and figure legend of Fig. 1.

Q3) Based on data presented in Fig. 2f, the authors hypothesize that there may be unidentified OM porins responsible for DDVA uptake at high concentrations. Is it also possible that passive diffusion of DDVA across the outer membrane could occur at 1 mM and 5 mM concentrations? DDVA is relatively hydrophilic and ionized at physiological pH due to the presence of two carboxyl groups. Because the cellular membrane is composed of a lipid bilayer, it is in general thought that ionized aromatic acids rarely pass through the membrane. Therefore, only a minor portion of DDVA may be passed through the membrane by diffusion.

Reference: Mori, K., Niinuma, K., Fujita, M., Kamimura, N. & Masai, E. DdvK, a novel major facilitator superfamily transporter essential for 5,5'-dehydrodivanillate uptake by *Sphingobium* sp. strain SYK-6. *Appl Environ Microbiol* **84**, doi:10.1128/AEM.01314-18 (2018).

Q4) On lines 190-192, the authors state that the combination of V42A and T43A mutations in the TonB box of DdvT decrease DDVA uptake “to levels comparable with Δ ddvT”. However, in Fig. 3c no tests are presented for uptake rates of the TonB point mutations against Δ ddvT. Instead, all comparisons are made against Δ ddcT+ddvT, which is not the comparison discussed in the text. Either or both comparison(s) could be justified, but please be consistent in the text and figure for the presented comparison(s).

Fig. 3c, L192: Thank you for your comment. We have added a statistically significant difference in DDVA uptake between Δ ddvT and Δ ddvT^{V42A-T43A} in Fig. 3c. When the TonB box mutants were compared to Δ ddvT + ddvT, we have clearly shown this in the text (L192).

Q5) The conclusion is presented that over-expression of DdvT “is an effective means for improving the efficiency of metabolite production...” (lines 261-267) and contributes to “improved efficiency of platform chemical production” (abstract, line 33). However, only fold-changes as compared to an empty vector control are presented for yield (1.3-fold increase), and “amount converted” (also 1.3-fold yield). It is difficult to understand how these values were derived from examining Fig. 6 alone, where it appears all of the DDVA was converted in both the empty-vector and DdvT over-expression strain. It seems the rate of PDC production may be the most improved metric, but this is not discussed. To support your claims, please provide a corresponding T/R/Y value in addition to the fold-changes presented in the text.

L33, L107, L277: As described in the text (L265-268), these fold-changes are a comparison of the ddvT-expressing *ligI* mutant with the *ligI* mutant harboring a vector (control) at 20 h after the cultivation. As pointed out by the reviewer, the phrase “improving the efficiency of metabolite production” is ambiguous. We corrected this phrase as “improving the **rate** of metabolite production” basically throughout the text (L33, L107, L277).

Q6) Minor edits:

a. Please define or provide a reference for “SEMP” in the methods (line 286) and Fig. 1 legend.

L289, L616: We have added a Reference (no. 36).

b. Please define the test used to compute p-values presented in Fig. 1 (lines 598-599).

L626-627: We have added the statistical analysis method.

c. In Fig. 2, what do the error bars represent? Please include in figure legend (p. 34-35).

L654-655: We have added an explanation of the error bars.

To Reviewer #2

Q1) In the abstract the authors claim uptake of aromatic compounds in TBDRs was previously unknown; however, this 2010 paper suggests the role of TBDRs in *Sphingomonas wittichii* RW1, specifically as transporters of aromatic hydrocarbons and/or their breakdown products. Genome Sequence of the Dioxin-Mineralizing Bacterium *Sphingomonas wittichii* RW1

Todd R. Miller, Arthur L. Delcher, Steven L. Salzberg, Elizabeth Saunders, John C. Detter, Rolf U. Halden. DOI: 10.1128/JB.01030-10

The authors later speak of upregulation of TBDR-like genes in *S. wittichii* RW1 and *Pseudomonas* strains, but the way the manuscript is presented, they are overstating the novelty of their findings.

L45-48: The paper provided by the reviewer only mentioned the possibility that TBDRs are involved in the uptake of aromatic hydrocarbons and/or their breakdown products based on the close localization of TBDR genes and predicted catabolic enzyme genes. Until now, the involvement of TBDR in aromatic compound uptake has been predicted from the viewpoint of gene locus and inducibility, however, there has been no experimental proof. Our paper is no doubt the first report experimentally demonstrating the involvement of the TonB system in the outer membrane transport and catabolism of an aromatic compound.

In this connection, since some siderophores transported by TBDR contain aromatic rings in the structure, we have rephrased as “Although there are examples where the TonB system transports certain siderophores containing aromatic groups, its involvement in **the uptake and catabolism of aromatic compounds** was previously unknown (L45-48).”

Q2) There is no clear explanation of the potential benefits of developing this system as a platform for lignin-derived compounds.

We think that the potential benefits are documented in L91-94 and L272-275.

Q3) Parts of the manuscript are overly speculative:

Page 12, lines 208-210: just because TBDR genes are upregulated by growth on lignin-derived compounds does not that those genes take up these compounds. This reviewer has unpublished data on another system refuting such a claim.

Here we only mentioned the possibility that TBDRs other than DdvT may also be involved in the uptake of lignin-derived aromatic compounds. This prediction is equivalent discussion to previous studies of *S. wittichii* RW1 and a *Pseudomonas* strain that predicted the possible involvement of TBDRs in the uptake of aromatic compounds based on the induction profiles of specific TBDR genes.

Pages 13-15: the whole section on component genes of the TonB complex necessary for DDVA uptake is poorly executed and overly speculative.

The conclusions in this section are logically derived from the results obtained and do not appear to be overly speculative.

Q4) The manuscript would be strengthened by routine assays in the TBDR field: purify the protein, perform binding studies, etc.

We agree with the comment. In the future, we need to purify the protein and perform binding assays (e.g. ITC) to enable the detailed biochemical characterization of the TonB system for DDVA uptake.

Q5) Figure 4: Although not discussed at all in the manuscript (so why is this included as a figure?), the TonB5 mutant appears to convert DDVA faster than wildtype. What is going on here?

Fig. 4 is essential to understand the results of the identification of components of the TonB complex necessary for DDVA uptake. Although not desirable, this figure can be moved to the supplementary material if necessary. Unfortunately, the reason why $\Delta tonB5$'s DDVA conversion is faster than that of the wild type is not clear at present.

Q6) Figure 5: The schematic of the DDVA uptake pathway shown in Figure 5 does not accurately depict the proposed model of the TonB complex in the inner membrane of Gram-negative bacteria. The authors should reference the following article: Nature volume 538, pages 60–65 (06 October 2016) for a better understanding of the components of the TonB system. This may help in improving their schematic

Thank you for your suggestion. We have improved Figure 5 according to the paper by Celia et al. (2016) and the recently published paper by Celia et al. (2019).

Q7) Finally, the inclusion of 14 supplementary figures and 4 tables is excessive and suggests that this manuscript is derived from a student's PhD thesis. Among the unnecessary supplementary figures are table 4, figure 2, and figure 4

In general, the primers used in the study should be described. Some leading journals require that they appear in the text. Because there is no space limitation in the Supplemental material, Figs. S2 and 4 should also be demonstrated as required research materials and analysis data, respectively.

To Reviewer #3

Q1) The manuscript is overall suitable for publication in Communications Biology. I would however encourage the authors to include a loading control for western blots shown in figures 2c, S6 and S9.

Figures 2c, S6, and S9; L415-417; 641-642: Thank you for pointing out. We performed ponceau S staining and showed loading controls in Figures 2c, S6, and S9.

REVIEWERS' COMMENTS:

Reviewer #1 (Remarks to the Author):

The authors have addressed my comments. I recommend this for publication now.

Reviewer #3 (Remarks to the Author):

The authors fulfill my request about including loading controls to their western blots. The manuscript is now suitable for publication in Communications Biology.

Response to reviewer's comments

There were no comments from reviewers in this revision. We appreciate the helpful and vital suggestions from the reviewers and the editor for our manuscript COMMSBIO-19-1060.